# Innovative practice of sustainable digital signal processing education

**Zhibo Xie** [ID]**\*, Zhongjie Zhu, Yuer Wang, Yongqiang Bai, Ruihua Zhang**

College of Information and Intelligence Engineering, Zhejiang Wanli University, Ningbo, China

\* xiezhibo@zwu.edu.cn

## Abstract

The Digital Signal Processing (DSP) course serves as a core curriculum for electronic information engineering majors. Traditional DSP instruction predominantly employs a teacher-centered lecture format supplemented by simulation experiments, which fails to engage student interest or foster proactive learning, ultimately resulting in suboptimal educational outcomes. This study integrates the BOPPPS instructional framework with Collaborative Inquiry-Based Learning (CIBL) models while designing simulation-hardware experiments that seamlessly bridge virtual and hardware-based environments. By incorporating Sustainable Development Goals (SDGs)-oriented engineering case studies into Project-Based Learning (PBL) content, this approach effectively stimulates student motivation, cultivates practical skills, and enhances problem-solving capabilities. The effectiveness of these pedagogical innovations is evaluated through academic performance analysis, student surveys, and focused interviews, enabling continuous optimization of teaching plans and content. Research findings demonstrate that integrating SDG-related engineering projects significantly boosts student motivation and professional confidence. Furthermore, the BOPPPS model combined with CIBL effectively improves learning efficiency and specialized competencies among students.

## 1. Introduction

In recent years, alongside rapid economic development, natural resources have been overexploited, leading to a deteriorating ecological environment and frequent occurrences of extreme weather events [1]. Critical pressures on ecosystems such as resource depletion, climate runaway, and biodiversity loss are compelling a transition toward sustainable development and the establishment of a global governance framework [2]. The 2015 Paris Agreement and the European Union's 2023 enactment of the New Battery Regulation and the Zero-Deforestation Regulation exemplify this shift [3]. Sustainable development has emerged as one of the most critical challenges requiring urgent understanding and action [4].

**Data availability statement:** All relevant data are within the paper and its Supporting Information files.

**Funding:** This research was funded by National First-Class First-Class Undergraduate Course - Digital Signal Processing, the second batch of higher education reform projects under Zhejiang Province's 14th Five-Year Plan (grant number JGBA2024410) and the second batch of regular provincial-level teaching reform projects for graduate education under Zhejiang Province's 14th Five-Year Plan (grant number JGCG2024365).

**Competing interests:** NO authors have competing interests.

**Abbreviations:** DSP, Digital Signal Processing; BOPPPS, Bridge-in, Objective, Pre-assessment, Participatory Learning, Post-assessment, and Summary; CIBL, Collaborative Inquiry-Based Learning; SDGs, Sustainable Development Goals; PBL, Project-Based Learning; VP, Virtual- Physical; DFT, Discrete Fourier Transform; FFT, Fast Fourier Transform; IIR, Infinite Impulse Response; FIR, Finite Impulse Response.

DSP serves as an "invisible engine" for sustainable development. In the energy sector, DSP addresses grid instability caused by the intermittency of renewable energy sources. For instance, the Fast Fourier Transform (FFT) algorithm enables power fluctuation prediction and dynamic adjustment of energy storage systems. Germany's E.ON grid company reduced wind curtailment from 8% to 2% through DSP applications, while Google's data centers achieved a 40% reduction in cooling energy consumption using DSP-optimized systems [5]. According to the International Renewable Energy Agency (IRENA), DSP technologies have cut global renewable energy grid integration costs by 25% [6]. Leveraging data-driven optimization capabilities, DSP provides a technological lever for sustainability across energy, environment, agriculture, and urban sectors [7]. By optimizing resource allocation, minimizing environmental footprints, and enhancing social well-being, DSP bridges technological innovation with sustainable development. Looking ahead, its integration with 5G, the Internet of Things (IoT), and artificial intelligence (AI) will further unlock its potential, driving humanity toward a sustainable model characterized by "zero waste, high efficiency, and low emissions" [8].

Consequently, DSP education has become increasingly vital. However, traditional DSP instruction—centered on classroom lectures and MATLAB-based verification experiments—often yields suboptimal outcomes, particularly in applied higher education institutions where students face greater challenges due to weaker mathematical foundations [9,10]. Effective DSP education must not only impart theoretical knowledge but also familiarize students with implementation workflows and real-world engineering applications [11–13]. This study aims to develop an innovative teaching methodology by integrating the BOPPPS instructional framework with collaborative inquiry-based learning. The approach incorporates blended online-offline learning platforms, virtual-physical experimental environments, and sustainability-themed engineering projects to enhance student engagement and revitalize pedagogy. Below is an overview of relevant literature.

## 2. Theoretical framework

### 2.1. BOPPPS

The BOPPPS model was initially proposed by Douglas Kerr from the University of British Columbia (UBC) in Canada in 1978 [14]. Originally, the model was primarily utilized for teacher skill training. It was created by the Instructional Skills Workshop (ISW) in British Columbia, Canada, to meet the province's teacher certification requirements [15]. During the training process, intensive and focused training is provided to enhance teachers' instructional skills and the effectiveness of their teaching. Nowadays, the BOPPPS model has been widely applied in over 100 universities and industrial training institutions across more than 33 countries worldwide [15]. The BOPPPS model is a novel, education-goal-oriented, and student-centered teaching approach. Rooted in constructivist and communicative language teaching theories, it is renowned for its effective instructional design. It is a closed-loop teaching process model that emphasizes student participation and feedback.

The model consists of six components: Bridge-in (B), Objective(O), Pre-assessment (P), Participatory Learning (P), Post-assessment (P), and Summary (S) [16]. In the Bridge-in phase, interest is captivated and the theme of the current lesson is introduced by employing methods such as storytelling and posing questions. The Objective phase involves clearly presenting the learning objectives to students, aiding them in comprehending the core content and skills to be mastered in this lesson, typically encompassing three aspects: knowledge, literacy, and skills. During the Pre-assessment phase, prior to formal instruction, students' existing knowledge is evaluated through methods like question-and-answer sessions and quizzes, thereby enabling adjustments to the teaching difficulty level. The Participatory Learning phase entails designing interactive activities, such as individual presentations, group discussions, and case analyses, to actively engage students in the construction of knowledge, serving as a crucial means of fostering their proactive learning abilities. In the Post-assessment phase, teaching effectiveness is evaluated through quizzes and questions. Finally, the Summary phase involves summarizing the content of the current lesson, consolidating key points, and introducing the content of the next lesson. All activities are centered around achieving students' learning objectives, with Participatory Learning serving as the core element. The bridge-in aims to capture students' attention and link learning content with their interests. Learning objectives are set to ensure the clarity of what students should accomplish after completing their studies. Pre-assessment involves testing students to analyze their learning status and prepare for subsequent teaching adjustments. Participatory learning comprises a range of engaging teaching activities that form the backbone of classroom instruction. Post assessment checks whether students have met their learning goals, and the summary reviews what they have learned. **Fig 1** is the BOPPPS model.

## 2.2. Project based learning

Project-based learning (PBL)is one of the primary methods in engineering education. It integrates practical application projects into classroom teaching content and generally includes stages such as project design, collaborative inquiry, iterative refinement of outcomes, and presentation and evaluation [17]. It plays a crucial role in fostering innovative thinking, enhancing the ability to integrate knowledge, and improving professional competence [18,19]. Through project-based

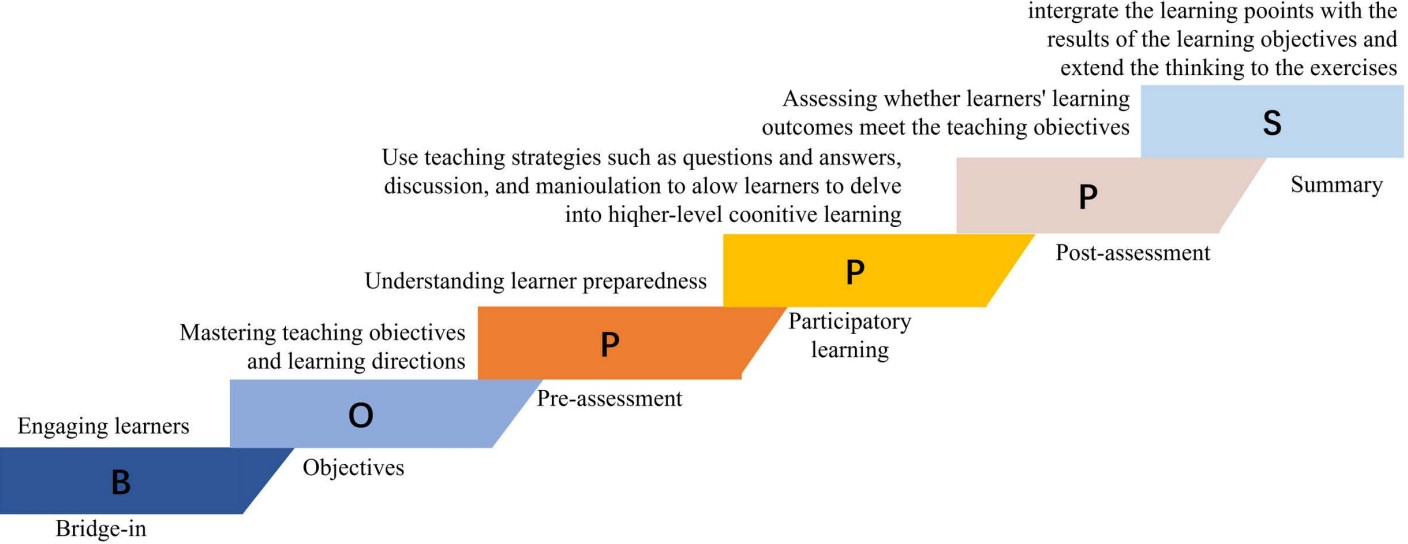

**Fig 1. BOPPPS model with six components: Bridge-in, objectives, pre-assessment, participatory learning, post-assessment, and summary.**

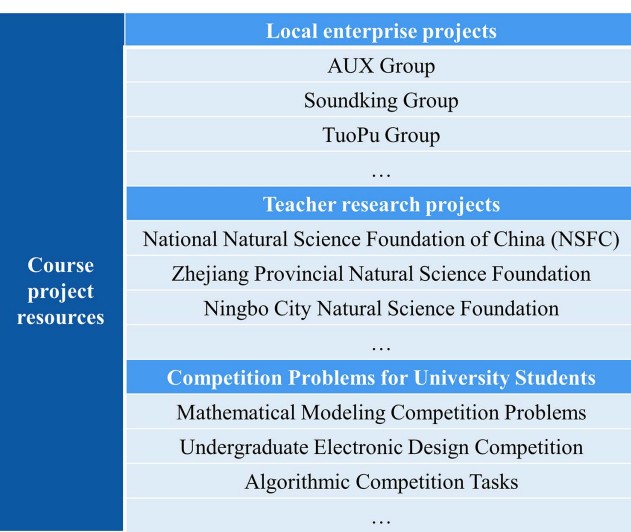

exercises, students can not only gain a better mastery of theoretical knowledge and integrate theory with practice more effectively, but also get familiarized with work scenarios in advance and cultivate their professional abilities.

PBL is based on a project as the main axis of the learning process, which guides students through the development of a subject by posing a challenge that cannot be solved solely by applying superficial knowledge [20]. The basic idea of PBL is to place students in real situations that require them to analyze, design, implement, and evaluate projects that have real applications, beyond the classroom examples. In this way, students will be able to apply what they have learned to solve problems in their future professional activities [21]. In this way, students will be able to apply what they have learned to solve problems in their future professional activities [22]. PBL has been applied as a teaching technique in several disciplines and learning levels [23]. It has been widely used in higher education levels.

Rooted in the concept of sustainable development, the project content of digital signal processing covers the applications of digital signal processing technology in noise reduction, smart cities, energy conservation and emission reduction. The project sources include local enterprise projects, teachers' research projects, and competition problems for university students from previous years as shown in **Fig 2**. Teachers extract parts of the content from these projects and integrate them into the teaching materials to design corresponding experiments. By completing these experiments, students not only master theoretical knowledge but also enhance their professional practical skills.

## 2.3. Virtual-physical experiment

Experiments are an indispensable part of engineering courses. Through experiments, theoretical knowledge can be transformed into practical applications, which helps students gain a better grasp of theoretical concepts and also allows them to hone their hands-on skills and professional competencies via practical operation. Generally, experiments can be categorized into virtual simulation experiments and hardware experiments. Sometimes, due to the high difficulty level of hardware experiments or the lack of experimental conditions, such as when experiments are hazardous or the required equipment is unavailable for purchase, teachers will design virtual simulation experiments instead. The Blended Virtual Reality Learning integrates virtual technologies (e.g., simulations, Virtual Reality/Augmented Reality, digital twins) with physical environments (e.g., laboratories, workshops, real-world projects) to create immersive, interactive, and adaptive learning experiences [24]. This approach bridges the gap between theory and

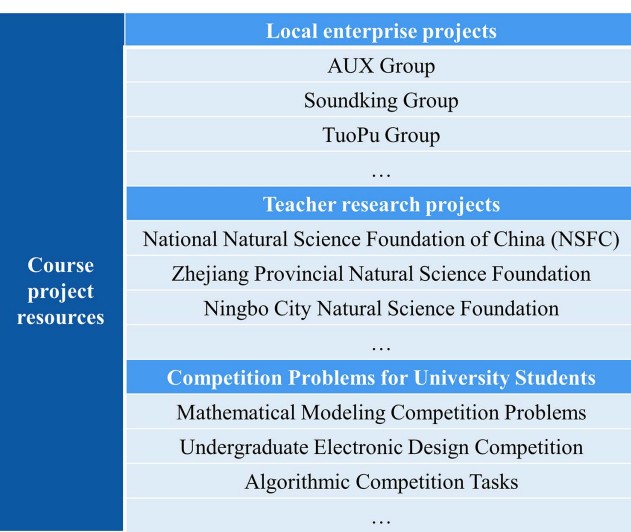

**Fig 2. Sources of course projects including local enterprise projects, teacher research projects, and competition problems.**

practice, enabling students to apply abstract knowledge to concrete contexts [25]. Its key advantages include enhancing student engagement, fostering more comprehensive skill development, and reducing equipment costs—as exemplified by initiatives like MIT's "Virtual Factory" and the University of Hong Kong's integration of VR walkthroughs with physical model-making [26]. This transformative teaching method amplifies traditional education through technological innovation.

In most universities, the experiments for DSP course typically employ MATLAB-based simulation verification. This is because MATLAB software is easy to operate, cost-effective, and user-friendly for students. Verification experiments mainly include convolution, FFT, Infinite Impulse Response (IIR) filter design, and Finite Impulse Response (FIR) filter design. However, after completing these experiments, students often remain unclear about how to apply these theoretical concepts in real-world scenarios, as MATLAB-based simulations differ significantly from actual implementations. A student might easily develop a MATLAB-based simulation program but struggle to implement a program based on a microcontroller. The primary distinction between application-oriented universities and research-oriented universities lies in their talent cultivation objectives. Application-oriented universities aim to cultivate individuals with strong practical skills. Therefore, in addition to traditional MATLAB-based simulation experiments, we have incorporated hardware experiments based on STM32 microcontrollers into the DSP course experiments to enhance students' engineering practical abilities.

### 2.4. Collaborative inquiry-based learning

CIBL is an instructional model that integrates inquiry-based learning with collaborative learning [27]. It emphasizes students working together in groups to propose questions, design inquiry plans, collect and analyze data, formulate conclusions, and share findings [28]. Through collaboration, students are able to learn from and inspire one another, jointly tackle problems, and thereby cultivate critical thinking, problem-solving skills, communication abilities, and teamwork spirit [29]. The core features of CIBL include problem-orientation and team collaboration, role differentiation and dynamic adjustment, resource integration and knowledge co-construction, as well as process-oriented reflection and iterative optimization [30]. CIBL is widely adopted in many universities as a method to develop students' skills in collaboration, problem-solving, and critical thinking [31]. CIBL appears to be effective in promoting learning motivation, learning engagement, classroom interaction, and higher-order thinking skills. CIBL is highly aligned with the core objectives and implementation pathways of engineering education philosophy, particularly in terms of being student-centered, practice-oriented, teamwork-focused, and committed to continuous improvement, where there exist close connections.

## 3. Materials and methods

This study constructs a sustainable knowledge chain. It designs teaching activities based on pedagogical experience to cultivate practical skills and problem-solving abilities. The effectiveness of teaching quality and learning outcomes is assessed by collecting and analyzing students' academic performance data, learning questionnaires, and conducting teaching interviews. Subsequently, the teaching model is reviewed and refined to develop a feasible teaching framework for sustainable DSP. The specific research objectives of this study are as follows:

1. To explore the feasibility of implementing the BOPPPS instructional framework and the CIBL model in the course of sustainable DSP.

2. To investigate the impact of adopting the BOPPPS instructional framework, the CIBL model, and virtual-real integrated experiments on students' professional knowledge and learning outcomes.

3. To determine students' perceptions and acceptance of the design and implementation of BOPPPS instructional framework and the CIBL model in the course of sustainable DSP.

## 3.1. Participants and context

The DSP course serves as a core professional course for the Electronics and Information Engineering major. This course is offered in the fifth semester at Zhejiang Wanli University in China, carrying a total of 3 credits and comprising 48 class hours. It adopts small-class teaching and is divided into two sections, accommodating a total of 78 students (60 males and 18 females). All students voluntarily participated in this experiment and signed informed consent forms. They were involved in the experiment from September 1, 2024, to December 31, 2024, with the option to voluntarily withdraw at any time during this period. Ethical approval for this study has been granted by the Ethics Committee of the College of Information and Intelligent Engineering, Zhejiang Wanli University (Approval Number: LLSC-2024–082001). All participants were provided with information about the study and gave written informed consent prior to participation. This study was conducted in accordance with the Declaration of Helsinki and all applicable ethical guidelines.

Zhejiang Wanli University is an applied university. Unlike research-oriented universities, applied universities are more closely aligned with regional or industrial demands, more dedicated to serving regional economic development, and place greater emphasis on cultivating students' practical abilities and vocational skills. Therefore, during the teaching process of the DSP course, greater emphasis is placed on the integration of theory with practice and the application of theoretical knowledge.

## 3.2. Course content design

The main content of the DSP course includes time-domain and frequency-domain analysis of discrete-time signals and systems, Discrete Fourier Transform (DFT), FFT, as well as the design and implementation of Infinite Impulse Response (IIR) and Finite Impulse Response (FIR) digital filters. The course is characterized by its strong theoretical orientation, with a large number of formulas and relatively few practical application examples. Most universities adopt a teaching approach that combines classroom theory instruction with MATLAB simulation experiments [27,32].

The course team adopts the philosophy of sustainable and innovative education, utilizing the BOPPPS model and project-driven teaching methods, while integrating practical application examples to restructure the course content. The content is then consolidated into four projects: classic audio effect processing, dial tone recognition, noise reduction, and image enhancement as shown in **Fig 3**. Each project is further subdivided into four modules, with each module corresponding to one class session and spanning three in-class hours. The primary theoretical knowledge for Project No. 1 encompasses the fundamental concepts, definitions, and time-domain as well as frequency-domain analyses of discrete-time signals and systems. The learning objectives are to gain an understanding of the general workflow and methodologies in signal processing, master frequency-domain analysis techniques, and comprehend the main technologies for time-domain signal processing through practical application examples of typical audio effects such as fast-forwarding, slow-motion playback, and voice alteration. The main content of Project No. 2 centers on the Discrete Fourier Transform (DFT) and the Fast Discrete Fourier Transform (FFT). The learning objectives are to master, through practical application examples, the physical significance of the Discrete Fourier Transform and FFT, as well as programming in C and MATLAB languages. Additionally, the aim is to comprehend the meanings of technical indicators in the field of signal processing, such as resolution and sampling duration. The main content of Project No. 3 focuses on the network structures of time-domain discrete systems and the design of Infinite Impulse Response (IIR) digital filters. The learning objectives are to be able to analyze the technical specifications of digital filters in the context of practical applications and to design IIR digital filters according to specific practical requirements. The content of Project No. 4 revolves around the network structures of time-domain discrete systems and the design of Finite Impulse Response (FIR) digital filters. The learning objectives are to master, through practical applications, the extraction of technical specifications for digital filters based on real-world application contexts, and to design FIR digital filters according to specific practical requirements. Each project is equipped with a project repository, and the application examples within the repository are sourced from teachers' research projects, past academic competition projects, and the needs of local enterprises, among others. The update rate of examples in the project repository is no less than 10% annually.

 

## Course content design based on PBL and VP experiment

| P1: Sound effect processing | P2: Dial tone recognition | P3: Noise reduction | P4: Image enhancement |
|---|---|---|---|
| Module:<br><br>•M1:Basic concepts and theories of time-discrete systems<br><br>•M2:Implementation of audio effect processing based on MATLAB<br><br>•M3:Implementation of audio effect processing based on STM32 microcontroller | Module:<br><br>•M1:Concepts and properties of FT, ZT, DFT<br><br>•M2:Principles and Applications of DFT<br><br>•M3:FFT simulation implementation<br><br>•M4:FFT implementation based on STM32 | Module:<br><br>•M1:Network structure of IIR filter<br><br>•M2:IIR filter design<br><br>•M3:Simulation Design and Application of IIR Filters<br><br>•M4:Implementation of IIR Filter Based on STM32 | Module:<br><br>•M1:Network structure of FIR filter<br><br>•M2:FIR filter design<br><br>•M3:Simulation Design and Application of FIR Filters<br><br>•M4:Implementation of FIR Filter Based on STM32 |

**Fig 3. Course content design based on PBL and virtual-physical experiments.** It includes 4 projects, each comprising 3-4 modules covering conceptual principles, MATLAB-based simulation, and STM32-based hardware implementation.

### 3.3. Teaching practice processes

The teaching implementation process comprises three stages: pre-class preparation, in-class application, and post-class improvement, as depicted in **Fig 4**. During the pre-class preparation stage, students engage in self-study using digital resources on online platforms.. In the in-class application stage, both teachers and students follow the BOPPPS model to accomplish project tasks. During the post-class improvement stage, students utilize online platforms to consolidate knowledge, engage in discussions, and participate in Q&A session. Additionally, they complete extension assignments in the open innovation laboratory. Meanwhile, teachers analyze course data through qualitative and quantitative methods to refine and improve teaching strategies.

**3.3.1. Pre-class preparation stage.** Teachers pre-record the theoretical knowledge lectures for each module and create digital lecture notes and quizzes. The theoretical knowledge lectures are generally presented in video format, while the digital lecture notes are in PPT form. The quizzes consist of objective questions such as multiple-choice or true/false questions on the Rain Classroom platform, primarily targeting the core knowledge points of the theoretical content. Before class, teachers release the teaching resources on an online learning platform, enabling students to clearly understand the learning objectives for that session, self-study the relevant theoretical knowledge, and gauge their understanding through the quizzes. If students fail the quiz, they can review the materials multiple times until they achieve a passing score.

**3.3.2. In-class application stage.** The BOPPPS model and CIBL approach are utilized during the classroom teaching stage. The entire classroom teaching is divided into six stages: bridge-in, objectives, pre-assessment, participatory learning, post-assessment, and summary. In the bridge-in stage, the teacher utilizes videos or informational materials to stimulate students' interest in learning, students watch video, read news, think and analyze. Subsequently, in the objectives stage, the teacher introduces the learning objectives and tasks, analyzes them, and students are grouped for task decomposition. Every team analyzes the task and objective. During the pre-assessment stage, students complete a quiz, and the teacher adjusts the difficulty level of tasks based on the quiz results. Participatory learning represents the most crucial stage, where each group engages in cooperative inquiry-based learning. After mastering

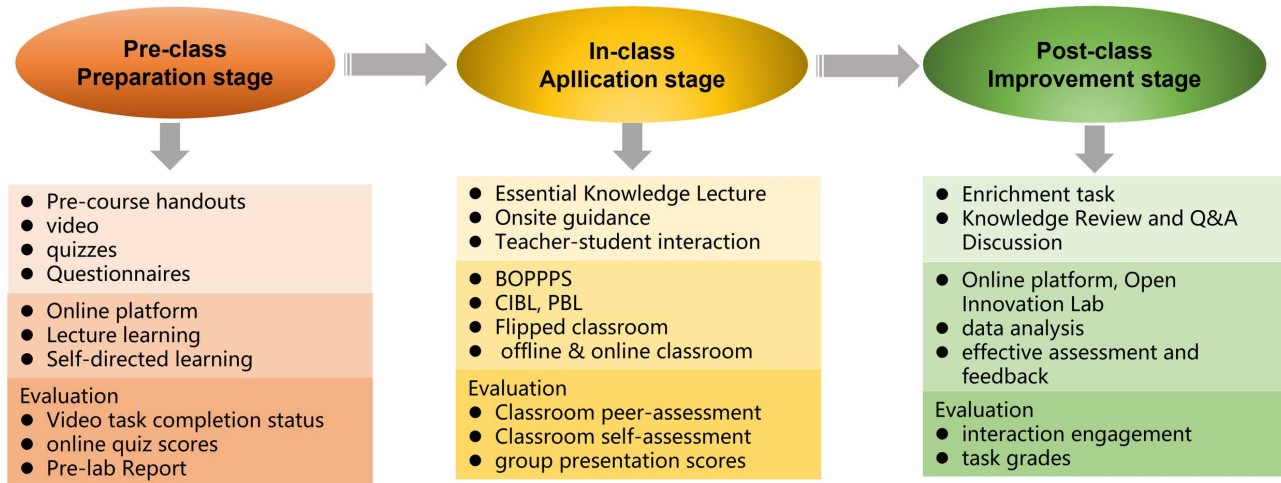

**Fig 4. Teaching implementation comprising pre-class preparation, in-class application, and post-class Improvement phases.** Each phase includes teaching resource preparation, primary instructional models, and evaluation.

theoretical knowledge, they complete virtual simulation experiment projects using MATLAB and then proceed to hardware experiments with STM32 microcontrollers, with the teacher providing assistance. The post-assessment stage serves as an evaluation of the outcomes of participatory learning, where each group completes a project and receives a grade based on the project's completion status. Finally, there is the summary stage. Students engage in self-assessment and peer assessment, while the teacher summarizes the task completion status, analyzes deficiencies, and implements continuous improvement measures. The whole implementation process is shown in **Fig 5**.

### 3.3.3. Post-class improvement stage.

In the post-class stage, students complete extended assignments in the open innovation laboratory, organize and consolidate their knowledge through interactive Q&A sessions on online platforms,

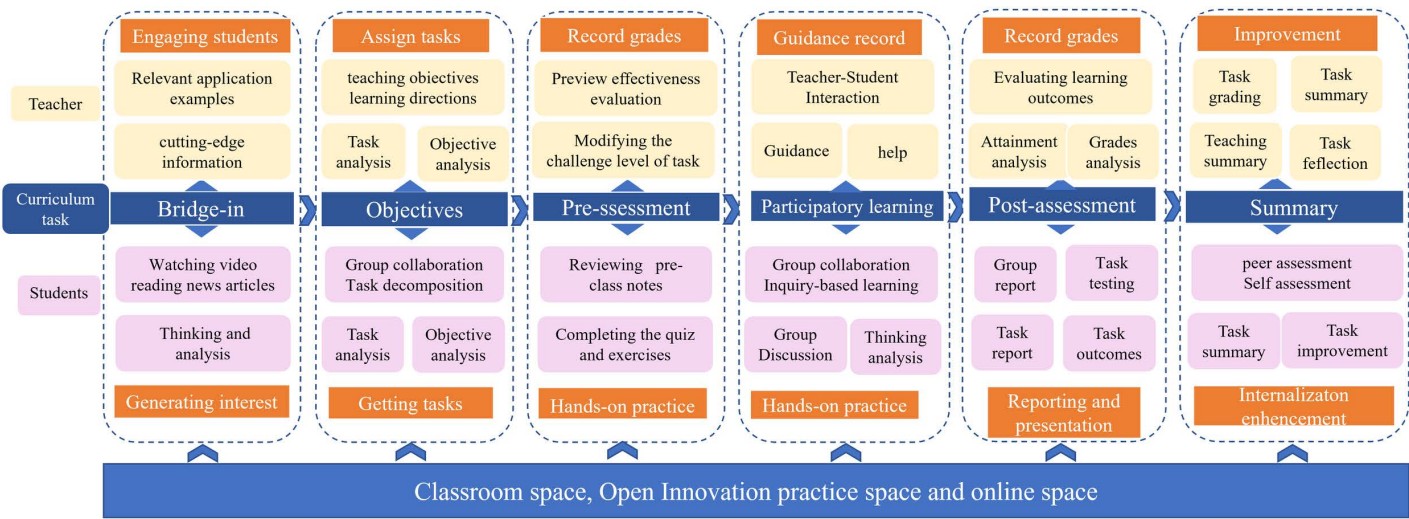

**Fig 5. In-class teaching implementation based on the BOPPPS model with six phases, integrating CIBL, PBL, and flipped class room approaches.** It details primary teacher-student behaviors and objectives for each phase.

and participate in feedback questionnaires. Meanwhile, teachers collect and analyze all data from the whole process, generating both quantitative and qualitative feedback on teaching effectiveness to refine and improve their instructional methods.

To evaluate the learning effectiveness of the BOPPPS model and CIBL approach, we conducted mixed-methods research combining qualitative and quantitative analyses. For the quantitative component, we utilized MOODLE and Rain Classroom platforms to assess students' learning capacities by reviewing their historical engagement data and providing real-time feedback. Both platforms maintained detailed quiz scores for each module, enabling us to collect and compare pre-test and post-test data.

Drawing on reference [33], we assessed students' acceptance of the innovative teaching model employed in this study from three dimensions: 1) motivation; 2) participation; 3) effectiveness. The study utilized a scale adapted from Jun-Scott Chen Hsieh's insights on flipped learning experiences [34] to measure these dimensions, as illustrated in **Table 1**. We designed a questionnaire to assess the course's impact. The questionnaire utilized a 5-point Likert scale, with scores ranging from "1" (strongly disagree) to "5" (strongly agree). Higher scores indicated a greater level of approval for the course. The questionnaire comprised 15 questions and was distributed to every student at both midterm and final exams. Regarding the qualitative research, we gathered qualitative feedback through interviews conducted at the end of the semester, asking questions such as "What have you learned in this course?" and "What are your learning experiences and suggestions regarding this course?".

**3.3.4. Data processing and analysis.** Before the midterm and final examinations of the semester, questionnaires and interviews were conducted on the teaching of the course (As shown in **Tables 1** and **2**), including learning satisfaction and learning suggestions. The data were collected and analyzed using paired sample *t*-tests, one-way ANOVA and qualitative analysis.

**Table 1. Survey on learning experience and learning satisfaction.**

| Motivation | Q1: I believe that this teaching model is a better way of learning. |
| --- | --- |
| | Q2: I like this teaching activity. |
| | Q3: I feel more motivated in this teaching model. |
| | Q4: I think the time and effort spent is worthwhile. |
| | Q5: I enjoy experiencing this teaching activity. |
| Participation | Q6: I become a more active learner. |
| | Q7: I invest more in the activities. |
| | Q8: I spend more time and effort on learning activities. |
| | Q9: I actively participate in group discussions. |
| | Q10: I find it easier to collaborate and interact with other students. |
| Effectiveness | Q11: I think it is a more effective and efficient way of learning. |
| | Q12: I participate more and invest more in course. |
| | Q13: I learn more and better. |
| | Q14: I believe it helps me gain a deeper understanding of the course content and skills. |
| | Q15: I think I am more independent in thinking and learning. |

**Table 2. Survey on learning suggestions.**

| S1: What have you learned in this course? |
| --- |
| S2: What are your learning experiences and recommendations in this course? |

## 4. Results and discussions

Firstly, we employed formative assessment data from pre-tests and post-tests conducted via the MOODLE and Rain Classroom platforms as the basis for evaluating learning outcomes, and analyzed the effectiveness of the innovative teaching approach. Secondly, we conducted both quantitative and qualitative analyses of students' learning experiences and satisfaction levels through questionnaires and focus group interviews. Subsequently, we tallied the competition awards earned by three consecutive cohorts of students upon completing the course to assess the impact of the innovative teaching method. Finally, we offered some suggestions regarding the application and promotion of the research methodology used in this study.

### 4.1. Learning effectiveness analysis

After students engaged in self-directed learning using pre-class materials, they were tested via the MOODLE or Rain Classroom platform system, which served as a pre-test to evaluate their baseline learning levels. This test was utilized as a pre-assessment to gauge students' learning standards. Following project-based classroom instruction and collaborative inquiry learning, a post-test was conducted again using the same platform system to examine the impact of the BOPPPS and CIBL model on learning outcomes (as shown in **Table 3**). M represents the mean and SD represents the standard deviation.

As can be seen from **Table 3**, the students' pre-test scores indicate that students did not read the pre-course materials or were unable to master the core knowledge of learning. When comparing the scores of the pre-test and post-test, it can be observed that the average score of the Unit 1 increased from 45 points to 81.72 points, marking an improvement of 36.72 points. The Cohen's d value was 7.12, with a 95% confidence interval of [6.85, 7.39], indicating a substantial effect of the teaching method. The average score of the Unit 2 rose from 47.30 points to 84.5 points, an increase of 37.2 points. The Cohen's d value was 4.41, and the confidence interval was [4.18, 4.649], which also implies that the teaching method has a large effect. The average score of the Unit 3 increased from 45.80 points to 84.173 points, showing an improvement of 42.93 points. The Co-hen's d value was 5, with a confidence interval of [4.75, 5.25], again demonstrating the noticeable effect of the teaching method. Finally, the average score of the Unit 4 increased from 42.50 points to 90.30 points, an increase of 47.80 points. The Cohen's d value was 5.22, and the confidence interval was [4.95, 5.49], further confirming the significant effect of the teaching method. In the post-test, the average scores of all project units were above 80 points, with the average score of Unit 4 exceeding 90 points. The research results showed that the post-test scores for all four projects are significantly higher than the pre-test scores by over 10 points, with a statistically significant difference. Moreover, the average scores demonstrate a gradual upward trend, indicating that the innovative curriculum indeed enhances students' learning effectiveness. Based on the results of descriptive statistical analysis of each student's scores, it was found that, overall, low-achieving and medium-achieving students demonstrated greater progress compared to high-achieving students. The primary reason is that through the innovative instructional design incorporating the BOPPPS

**Table 3. Analysis of a sample of 78 students in project units with paired sample t-test.** *N* represents the sample size, M denotes the mean, SD indicates the standard deviation, Cohen's d signifies the effect size, and the 95% confidence interval for d is provided.

| Project unit | *N* | Pre-test | | Post-test | | Paired difference | *Paired t-test* | Cohen's d | d |
|---|---|---|---|---|---|---|---|---|---|
| | | M | SD | M | SD | | | | |
| 1: Sound effect processing | 78 | 45.00 | 22.12 | 81.72 | 16.64 | 36.72 | −62.45*** | 7.12 | [6.85, 7.39] |
| 2: Dial tone recognition | 78 | 47.30 | 18.52 | 84.50 | 15.47 | 37.20 | −38.96 *** | 4.41 | [4.18, 4.64] |
| 3: Noise reduction | 78 | 45.80 | 16.13 | 88.73 | 14.42 | 42.93 | −44.4 0*** | 5.00 | [4.75, 5.25] |
| 4: Image enhancement | 78 | 42.50 | 18.37 | 90.30 | 13.63 | 47.80 | −44.85*** | 5.22 | [4.95, 5.49] |

***: $p < 0.001$.

model and CIBL approach, group discussions are integrated with teaching activities. Teachers encourage students to help one another within groups, allowing high-achieving students to lead the learning process and motivate low- and medium-achieving students, thereby enabling the latter group to make greater progress.

## 4.2. Learning experience and learning satisfaction survey

**4.2.1. Analysis of the questionnaire on learning experience and learning satisfaction.** At the end of the semester, we conducted a questionnaire survey on learning cognition and learning satisfaction to understand learners' desires, feelings, and attitudes towards learning activities. The questionnaire comprises 15 questions, designed based on a 5-point Likert scale ranging from 1 to 5, representing "strongly disagree," "disagree," "neutral," "agree," and "strongly agree," respectively. All 78 students were invited to participate in the survey, consisting of 60 male and 18 female students. A total of 78 questionnaires were collected, achieving a response rate of 100%. Statistical analysis of the 78 questionnaires revealed that the questionnaire demonstrated good reliability, with a Cronbach's α of 0.918 (>0.8), and validity of 0.834 (>0.5), with a p-value of 0.000 (<0.05). Specific survey results are presented in **Table 4**.

**Table 4** shows that "Q5: I enjoy experiencing this teaching activity." received the highest score, indicating that all students endorse this innovative teaching model, which effectively stimulates the learning interest of all students. This is followed by "Q7: I invest more in the activities." and "Q11: I think it is a more effective and efficient way of learning." suggesting that students have spent more time in learning the course and achieved notably significant gains throughout their studies. The mean of motivation is 4.30. The mean of participation is 4.298. The mean of effectiveness is 4.214. The mean of all questions is 4.27. Overall, students' general perception of learning and their satisfaction are positive.

**4.2.2. Analysis of interviews on learning suggestions.** Student feedback stands as one of the most crucial criteria for evaluating teaching effectiveness. Particularly, through qualitative interview feedback, educators can not only gain insights into students' learning experiences and suggestions but also understand their learning situations and needs. This provides teachers with fresh perspectives for teaching reflection, enabling them to promptly adjust the teaching pace, methodologies, and course content. In this study, focus group interviews were conducted via special feature reports, and the specific feedback analysis is presented as follows.

**Table 4. Data analysis of the questionnaire survey. *N* represents the sample size, MIN denotes the minimum value, MAX indicates the maximum value, M stands for the mean, and SD signifies the standard deviation.**

| Dimension | Questions | *N* | MIN | MAX | M | SD |
|---|---|---|---|---|---|---|
| Motivation | Q1 | 78 | 3 | 5 | 4.10 | 0.37 |
| | Q2 | 78 | 3 | 5 | 4.09 | 0.49 |
| | Q3 | 78 | 3 | 5 | 4.12 | 0.48 |
| | Q4 | 78 | 3 | 5 | 4.18 | 0.58 |
| | Q5 | 78 | 5 | 5 | 5.00 | 0.00 |
| Participation | Q6 | 78 | 3 | 5 | 4.26 | 0.54 |
| | Q7 | 78 | 4 | 5 | 4.88 | 0.12 |
| | Q8 | 78 | 3 | 5 | 4.06 | 0.70 |
| | Q9 | 78 | 3 | 5 | 3.97 | 0.63 |
| | Q10 | 78 | 4 | 5 | 4.32 | 0.24 |
| Effectiveness | Q11 | 78 | 4 | 5 | 4.90 | 0.12 |
| | Q12 | 78 | 3 | 5 | 3.95 | 0.66 |
| | Q13 | 78 | 3 | 5 | 3.89 | 0.71 |
| | Q14 | 78 | 3 | 5 | 4.14 | 0.64 |
| | Q15 | 78 | 3 | 5 | 4.19 | 0.67 |

Suggestion 1: Incorporating sustainable development projects into the curriculum can enhance students' learning engagement. This study leverages practical application cases to stimulate students' interest and curiosity, as well as enhance their willingness to participate in the course. For instance, in the audio effects processing project, students were tasked with achieving effects such as fast-forwarding, slowing down, reversing, and voice modulation on a piece of music. This added an element of fun to the course and facilitated a better understanding of frequency domain concepts among students. In the dial tone recognition project, we encouraged two groups of students to test and evaluate each other's recognition results, fostering a sense of competition. For the noise reduction project, we incorporated examples of noise classification and its hazards to the environment and living organisms. Additionally, students were guided to obtain electrocardiogram (ECG) experimental data from the Massachusetts Institute of Technology (MIT), which not only heightened their sense of professional mission but also boosted their confidence in their field of study.

Suggestion 2: The synergistic innovation of the BOPPPS and CIBL models can promote students' proactive self-directed learning. This course integrates tedious theoretical knowledge with practical applications, making it easier for students to comprehend, reducing the learning difficulty, and transforming abstract theoretical concepts into tangible ones. Before class, students can assess their learning progress through knowledge lectures and quizzes on an online platform, and they have the flexibility to review the material repeatedly. During class, student-centered group collaborative inquiry-based learning encourages peers to assist and engage in discussions with one another, while teachers provide guidance. These approaches facilitate faster, clearer, and more engaging learning experiences for students, and additionally enhance their learning motivation and conceptual understanding. Virtual experiments based on MATLAB and hardware experiments based on STM32 microcontroller enable students to gain a better grasp of theoretical knowledge in DSP. Students prefer this teaching method over traditional lecture-based approaches. The aforementioned analysis indicates that both the BOPPPS teaching model and the project-driven, collaborative inquiry-based learning mode are highly effective. Students generally favor integrating course content with practical examples, staying updated on cutting-edge technologies in the signal processing field, understanding professional requirements in signal processing, developing practical skills, and swiftly mastering key learning points in a relaxed and enjoyable learning environment.

### 4.3. Student awards

The sustainable and innovative teaching model has sparked students' interest in this course, bolstered their professional confidence, and enhanced their practical abilities. During or after studying this course, an increasing number of students have participated in various competitions and garnered numerous accolades. Taking the National Undergraduate Electronic Design Contest as an example, which is one of the most authoritative contests closely aligned with this course in China, the average award-winning rate among students from applied universities does not exceed 20%. The course team has compiled statistics on the award-winning rates of students majoring in Electronic Information Engineering at our university over the past three years, as shown in **Table 5**. From **Table 5**, it can be observed that the award-winning rate was 15% in 2022, slightly below the average level of applied universities; it rose significantly to 23.68% in 2023, slightly surpassing the average; and in 2024, it reached 38.46%, far exceeding the average. This demonstrates that the innovative reforms in the course have a notable impact on enhancing students' professional capabilities.

**Table 5. Student awards in national undergraduate electronic design contest in the past three years.**

| Year | Total Number of Students | Number of Award-Winning Students | Award Rate (%) |
|------|--------------------------|----------------------------------|----------------|
| 2024 | 78 | 30 | 38.46 |
| 2023 | 76 | 18 | 23.68 |
| 2022 | 80 | 12 | 15.00 |

## 4.4. Suggestions and discussions

This innovative teaching method requires a substantial amount of resources, both online and offline, primarily including recording instructional videos for theoretical lectures, creating test question banks (for pre-tests and post-tests), establishing an engineering project case library (which is regularly updated), and designing experimental content that combines virtual and hardware elements. All these tasks demand a significant pre-class investment of effort from the teachers in the research group. Additionally, it necessitates the provision of corresponding platforms by the school, such as the Rain Classroom software purchased by the authors' university and the self-developed MOODLE online learning platform managed by the university's information center. This innovative teaching method requires a substantial amount of resources, both online and offline, primarily including recording instructional videos for theoretical lectures, creating test question banks (for pre-tests and post-tests), establishing an engineering project case library (which is regularly updated), and designing experimental content that combines virtual and hardware elements. All these tasks demand a significant pre-class investment of effort from the teachers in the research group. Additionally, it necessitates the provision of corresponding platforms by the school, such as the Rain Classroom software purchased by the authors' university and the self-developed MOODLE online learning platform managed by the university's information center.

Furthermore, while this method has been applied to the Digital Signal Processing course in this paper, it can also be extended and applied to other engineering practice-oriented courses. This approach is particularly well-suited for courses with an engineering background and a strong emphasis on practicality, such as EDA Technology and Digital Electronics Technology courses, where it is expected to yield favorable results as well.

## 5. Conclusions

This study employed the BOPPPS instructional model and CIBL model for sustainable DSP education. The main conclusions are as follows:

1. The post-test scores for each unit are more than 20 points higher than the pre-test scores, indicating that incorporating sustainable project cases into the teaching content can stimulate students' learning interest.

2. The post-test scores for each project of every student were higher than the pre-test scores, demonstrating that the BOPPPS instructional model and CIBL model can stimulate students' learning interest and improve teaching effectiveness.

3. Students provided highly positive feedback on the BOPPPS instructional model and collaborative inquiry-based learning model for sustainable DSP education. They generally appreciated the integration of course content with practical applications and the ability to quickly engage in learning scenarios through group work, which effectively improved the quality of learning outcomes. The overall average scores for learning experience and learning satisfaction both exceeded 4.2.

4. Qualitative feedback from students indicated that the use of the BOPPPS instructional model and collaborative inquiry-based learning model, combined with the integration of sustainable development projects, enhanced students' interest and participation in teaching activities. In particular, during group work, students with varying academic performances could assist and learn from one another.

Overall, the BOPPPS and CIBL models, which integrate engineering cases, cutting-edge technologies, and sustainable development applications into practical projects for teaching and learning, represent an innovative and effective teaching approach. They also provide a reference for diversified teaching strategies in related disciplines.

## Supporting information

**S1 File. Aggregated response information from all 78 students to the questionnaire.**
(XLSX)

**S2 File. Student award information.**
(XLSX)

**S3 File. Test record.**
(XLSX)

## Acknowledgments

We would like to express our special thanks to the students who participated in this study. We are very grateful to them.

## Author contributions

**Conceptualization:** Zhibo Xie, Zhongjie Zhu.

**Data curation:** Yuer Wang.

**Formal analysis:** Ruihua Zhang.

**Funding acquisition:** Zhongjie Zhu.

**Investigation:** Yuer Wang.

**Resources:** Zhibo Xie, Ruihua Zhang.

**Software:** Yongqiang Bai.

**Supervision:** Ruihua Zhang.

**Validation:** Yuer Wang.

**Visualization:** Yongqiang Bai.

**Writing – original draft:** Zhibo Xie.

**Writing – review & editing:** Zhibo Xie.

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
