## [Decision Letter · Decision Letter 0]

17 Nov 2025

Dear Dr. xie,

We look forward to receiving your revised manuscript.

Kind regards,

Fatih Uysal, Ph.D.

Academic Editor

PLOS ONE

Journal Requirements:

4. We note that Figure 3 in your submission contain copyrighted images. All PLOS content is published under the Creative Commons Attribution License (CC BY 4.0), which means that the manuscript, images, and Supporting Information files will be freely available online, and any third party is permitted to access, download, copy, distribute, and use these materials in any way, even commercially, with proper attribution. For more information, see our copyright guidelines: http://journals.plos.org/plosone/s/licenses-and-copyright.

1. You may seek permission from the original copyright holder of Figure 3 to publish the content specifically under the CC BY 4.0 license.

5. We note that Figure 4 includes an image of a participant in the study.

If you are unable to obtain consent from the subject of the photograph, you will need to remove the figure and any other textual identifying information or case descriptions for this individual."

6. Please include captions for your Supporting Information files at the end of your manuscript, and update any in-text citations to match accordingly. Please see our Supporting Information guidelines for more information: http://journals.plos.org/plosone/s/supporting-information .

7. Thank you for providing your underlying data as Supporting Information.

We note that the data set contains text or data that is not in English. Please note that PLOS is an English-language publisher, so we require data sets to be provided in English as well. Please upload an English-language version of your data set.

This will also allow us to determine if your data follows PLOS standards per our Data Availability policy here: https://journals.plos.org/plosone/s/data-availability

8. We note that there is identifying data in the Supporting Information file “student award information(english ).xlsx”. Due to the inclusion of these potentially identifying data, we have removed this file from your file inventory. Prior to sharing human research participant data, authors should consult with an ethics committee to ensure data are shared in accordance with participant consent and all applicable local laws.

-Location data

Additional Editor Comments:

Revise your paper according to the referee comments.

Reviewers' comments:

Reviewer's Responses to Questions

**Comments to the Author**

1. Is the manuscript technically sound, and do the data support the conclusions?

Reviewer #1: Partly

Reviewer #2: Yes

2. Has the statistical analysis been performed appropriately and rigorously?

Reviewer #1: No

Reviewer #2: Yes

3. Have the authors made all data underlying the findings in their manuscript fully available?

Reviewer #1: Yes

Reviewer #2: Yes

4. Is the manuscript presented in an intelligible fashion and written in standard English?

Reviewer #1: No

Reviewer #2: Yes

Reviewer #1: 1. This instructional strategy is innovative and has the potential for broader application. While the paper is a valuable contribution, some revisions are necessary before it can be considered for publication.

2. Please provide a brief introduction to the BOPPPS model earlier in the article.

3. Please provide evidence of the reliability or maybe validity for the survey scale used in the study.

4. Please provide more detailed statistical values, such as the paired Cohen’s d and 95% CIs.

5. The paper would be strengthened by including the explicit results from the qualitative analysis, which the authors state was conducted in Section 3.3.4.

6. A thorough language and technical edit is required prior to resubmission to address several errors in the text. For example: Reference 6 appears to be missing from the in-text citations, and the definition provided for the abbreviation VR is incorrect.

Reviewer #2: Overview

This study describes an effective and relevant approach to teaching Digital Signal Processing by integrating sustainability themes into the learning process. The combination of the BOPPPS model and CIBL framework promotes student participation, critical thinking, and understanding of the connection between engineering education and sustainable development. The topic aligns with current trends in educational innovation and applied learning.

Major Strengths

• Clear and well-motivated instructional design linking DSP education with sustainability goals.

• Ethical approval and student consent are clearly documented.

• Balanced use of qualitative and quantitative assessment methods.

• Open sharing of anonymized data and transparent research procedures.

• Logical structure and clear presentation of results.

Points for Improvement (Minor Revisions)

1. Provide basic demographic information such as sample size, gender distribution, and response rate for clarity.

2. Add statistical details such as confidence intervals or effect sizes to show the significance of improvements.

3. Improve figure and table captions so that they are self-explanatory.

4. Discuss briefly how the proposed model can be applied to other engineering or online teaching contexts.

5. Review the manuscript for minor grammatical corrections and consistent terminology.

Ethics and Data Availability

Ethical approval (LLSC-2024-082001) and written consent were properly obtained. All data are available in the supplementary materials and meet the PLOS ONE open-data requirements. No ethical concerns were found.

Overall Recommendation

Minor Revision. The paper is methodologically sound, ethically compliant, and presents a valuable contribution to sustainable engineering education. Only minor additions to statistical reporting and formatting are needed before publication.

Reviewer Certification

This review was written independently and reflects my professional evaluation of the manuscript. It complies with PLOS ONE and COPE Ethical Guidelines for Peer Reviewers.

**Do you want your identity to be public for this peer review?** For information about this choice, including consent withdrawal, please see our Privacy Policy

Reviewer #1: **Yes:** Chih-Cheng Lin

Reviewer #2: **Yes:** Sara Goze

---

## [Author Response · Author response to Decision Letter 1]

8 Dec 2025

Response to Reviewers

Journal Requirements:

We have chected all our manuscripts, they all meet PLOS ONE's style requirements.

we have remove the funding-related text from the manuscript.

We have revised the information.

4. We note that Figure 3 in your submission contain copyrighted images. All PLOS content is published under the Creative Commons Attribution License (CC BY 4.0), which means that the manuscript, images, and Supporting Information files will be freely available online, and any third party is permitted to access, download, copy, distribute, and use these materials in any way, even commercially, with proper attribution. For more information, see our copyright guidelines: http://journals.plos.org/plosone/s/licenses-and-copyright.

we have removed the Figure 3.

5. We note that Figure 4 includes an image of a participant in the study.

If you are unable to obtain consent from the subject of the photograph, you will need to remove the figure and any other textual identifying information or case descriptions for this individual."

We have removed the figures and any other textual identifying information.

We have added a title to our supporting information.

7. Thank you for providing your underlying data as Supporting Information.

We note that the data set contains text or data that is not in English. Please note that PLOS is an English-language publisher, so we require data sets to be provided in English as well. Please upload an English-language version of your data set.

This will also allow us to determine if your data follows PLOS standards per our Data Availability policy here: https://journals.plos.org/plosone/s/data-availability

We have uploaded an English-language version of our data set.

8. We note that there is identifying data in the Supporting Information file “student award information(english ).xlsx”. Due to the inclusion of these potentially identifying data, we have removed this file from your file inventory. Prior to sharing human research participant data, authors should consult with an ethics committee to ensure data are shared in accordance with participant consent and all applicable local laws.

We have removed all personal information in the Supporting Information file “student award information(english ).xlsx” to ensure data are shared in accordance with participant consent and all applicable local laws.

-Location data

We have removed all personal characteristic-related information from the supporting information file named "Student Award Information".

Comments to the Author

1. Is the manuscript technically sound, and do the data support the conclusions?

Reviewer #1: Partly

Reviewer #2: Yes

2. Has the statistical analysis been performed appropriately and rigorously?

Reviewer #1: No

Reviewer #2: Yes

We have incorporated detailed statistical analysis content.

3. Have the authors made all data underlying the findings in their manuscript fully available?

Reviewer #1: Yes

Reviewer #2: Yes

4. Is the manuscript presented in an intelligible fashion and written in standard English?

Reviewer #1: No

Reviewer #2: Yes

We have thoroughly revised the expressions in the paper to ensure the use of standard English.

5. Review Comments to the Author

Reviewer #1:

1. This instructional strategy is innovative and has the potential for broader application. While the paper is a valuable contribution, some revisions are necessary before it can be considered for publication.

We have made conscientious revisions based on the reviewer's comments.

2. Please provide a brief introduction to the BOPPPS model earlier in the article.

We have added a brief introduction to the BOPPPS model in 2.1.

3. Please provide evidence of the reliability or maybe validity for the survey scale used in the study.

We have added reliability information in 4.2.

4. Please provide more detailed statistical values, such as the paired Cohen’s d and 95% CIs.

We have added paired Cohen’s d and 95% CIs information in table 3.

5. The paper would be strengthened by including the explicit results from the qualitative analysis, which the authors state was conducted in Section 3.3.4.

The original paper did include content on qualitative analysis, but it was not presented clearly enough. Therefore, we have reorganized the subsections in Chapter 4 of the paper, with Section 4.2 dedicated to qualitative analysis.

6. A thorough language and technical edit is required prior to resubmission to address several errors in the text. For example: Reference 6 appears to be missing from the in-text citations, and the definition provided for the abbreviation VR is incorrect.

The indexing error in Reference 6 has been corrected, and all spelling mistakes and technical errors in the paper have been rectified.

Reviewer #2: Overview

This study describes an effective and relevant approach to teaching Digital Signal Processing by integrating sustainability themes into the learning process. The combination of the BOPPPS model and CIBL framework promotes student participation, critical thinking, and understanding of the connection between engineering education and sustainable development. The topic aligns with current trends in educational innovation and applied learning.

Major Strengths

• Clear and well-motivated instructional design linking DSP education with sustainability goals.

• Ethical approval and student consent are clearly documented.

• Balanced use of qualitative and quantitative assessment methods.

• Open sharing of anonymized data and transparent research procedures.

• Logical structure and clear presentation of results.

Points for Improvement (Minor Revisions)

1. Provide basic demographic information such as sample size, gender distribution, and response rate for clarity.

We have added the sample size, gender distribution, and response rate in 4.2.1.

2. Add statistical details such as confidence intervals or effect sizes to show the significance of improvements.

We have added confidence intervals and effect sizes in table 3.

3. Improve figure and table captions so that they are self-explanatory.

We have improve the information in all figures and table titles.

4. Discuss briefly how the proposed model can be applied to other engineering or online teaching contexts.

Section 4.4 has been supplemented with a brief discussion on the application of the proposed solution in other engineering-related courses.

5. Review the manuscript for minor grammatical corrections and consistent terminology.

We have corrected the grammatical errors and inconsistencies in terminology usage.

---

## [Editor Report · Decision Letter 1]

28 Dec 2025

Innovative Practice of Sustainable Digital Signal Processing Education

PONE-D-25-40023R1

Dear Dr. xie,

We’re pleased to inform you that your manuscript has been judged scientifically suitable for publication and will be formally accepted for publication once it meets all outstanding technical requirements.

Kind regards,

Fatih Uysal, Ph.D.

Academic Editor

PLOS One

Additional Editor Comments (optional):

Thank you for the revision. The revisions to the article and the responses to the reviewers’ comments are satisfactory, and the article has been accepted.
---

## [Editor Report · Acceptance letter]

PONE-D-25-40023R1

PLOS One

Dear Dr. xie,

I'm pleased to inform you that your manuscript has been deemed suitable for publication in PLOS One. Congratulations! Your manuscript is now being handed over to our production team.

Kind regards,

on behalf of

Dr. Fatih Uysal

Academic Editor

PLOS One